# Implementing an Agent-Based Modeling Approach for Protein Glycosylation in the Golgi Apparatus

Christian Jetschni and Peter Götz *

Bioprocess Engineering, Faculty of Life Sciences and Technology, Berliner Hochschule für Technik (BHT), 13347 Berlin, Germany; christian.jetschni@bht-berlin.de
* Correspondence: peter.goetz@bht-berlin.de

**Abstract:** Glycoproteins are involved in various significant biological processes and have critical biological functions in physiology and pathology by regulating biological activities and molecular signaling pathways. The variety of enzymes used in protein glycosylation and the wide range of diversity in the resulting glycoproteins pose a challenging task when attempting to simulate these processes in silico. This study aimed to establish and define the necessary structures to simulate the process of N-glycosylation in silico. In this article, we represent the process of glycosylation in the Golgi structure in an agent-based model with defined movement patterns and reaction rules between the associated proteins and enzymes acting as agents. The Golgi structure is converted into a grid consisting of $150 \times 400$ patches representing four compartments which contain a specific distribution of the fundamental enzymes contributing to the process of glycosylation. The interacting glycoproteins and membrane-bound enzymes are perceived as agents, with their own rules for movement, complex formation, biochemical reaction and dissociation. The resulting structures were saved into an XML-format, a mass spectrometry file and a GlycoWorkbench2-compatible file for visualization.

**Keywords:** glycosylation; Golgi apparatus; glycobiology; simulation; modeling; proteomics

## 1. Introduction

Glycosylation refers to a series of enzymatic reactions in which the glycosyl donors are bound to proteins, lipids or other functional groups of a molecule. Most of its processes occur either co-translationally or post-translationally in the cytosol, in the endoplasmic reticulum and in the Golgi apparatus. The resulting glycans make an important contribution to increasing the overall stability of glycoproteins, making simulation for the optimization and prediction of the parameters of glycosylation and the resulting glycans a promising approach to analyze and optimize the stability of protein pharmaceuticals. The strategy of N-linked glycosylation of proteins has been applied in modern drug development. For instance, the engineered N-linked glycosylation is used to stabilize recombinant proteins or peptides. Particularly, most of the short peptides in nature are not stable and can be easily degraded, with their half-lives being very short, limiting their clinical applications. The glycosylation of the Asn-X-Ser/Thr sequence is artificially engineered and fused in the recombinant genes of interest, leading to the expression of the glycosylated recombinant proteins, antibodies, peptides or fusion proteins. This can enhance the stability of proteins and peptides of interest extremely and increase their potential clinical uses [1,2]. Over the years, significant improvements in experimental analytical methods have led to the development of many software tools that aim to assist in the interpretation of the experimental data generated. Currently, mainly mass spectrometry (MS) and high-performance liquid chromatography (HPLC) are utilized due to their sensitivity [3]. Carbohydrates exhibit high structural diversity. Due to the difficulty of site-specifically controlling glycosylation at each of the various positions within a protein, understanding and engineering glycoproteins remains challenging [4].

In the field of biology, simulations are increasingly being used to investigate and understand complex processes. In order to be able to create and analyze simulations, the biological processes must be modeled. The representation of biological phenomena and processes by mathematical models allows us to reduce the complexity and to predict and understand the impact of changes within a system.

## 2. Theoretical Background

Glycoproteins are glycan-bearing proteins, and the amino acid to which the glycans are bound defines the type of glycosylation. There are several types of glycosylation, including N-linked glycosylation (N-glycosylation), O-linked glycosylation (O-glycosylation), C-mannosylation, phospho-glycosylation and glypiation [5]. The two major types of glycosylation, N-linked and O-linked (Figure 1), are both involved in the maintenance of proteins' conformation and activity, in protecting proteins from proteolytic degradation, and in the intracellular trafficking and secretion of proteins [6,7]. For N-glycosylation, the glycan is attached to an asparagine residue embedded in an Asn-X-Ser/Thr/Cys motif [8,9]. Glycans are among the most complex biological molecules found in nature due to their diverse and asymmetric type of branching. N-glycan groups play a key role in the folding, processing and secretion of proteins from the endoplasmic reticulum (ER) and the Golgi apparatus [6]. Ultimately the glycan signals whether a protein is correctly folded and can leave the ER to continue its maturation in the Golgi apparatus, or whether the protein is not correctly folded and is degraded [10]. N-glycosylation often occurs co-translationally, as the glycan is covalently linked to the nascent protein via an N-glycosidic bond with an asparagine residue within the primary protein sequence of Asn-X-Ser/Thr/Cys, where X is any amino acid except proline [9,10]. However, the sole presence in the amino acid sequence is not sufficient to predict the presence of a glycan, since not all Asn residues with the predicted consensus sequence are glycosylated. Only about two-thirds of the consensus sequences are glycosylated, and almost 90% of glycoproteins are N-glycosylated [11].

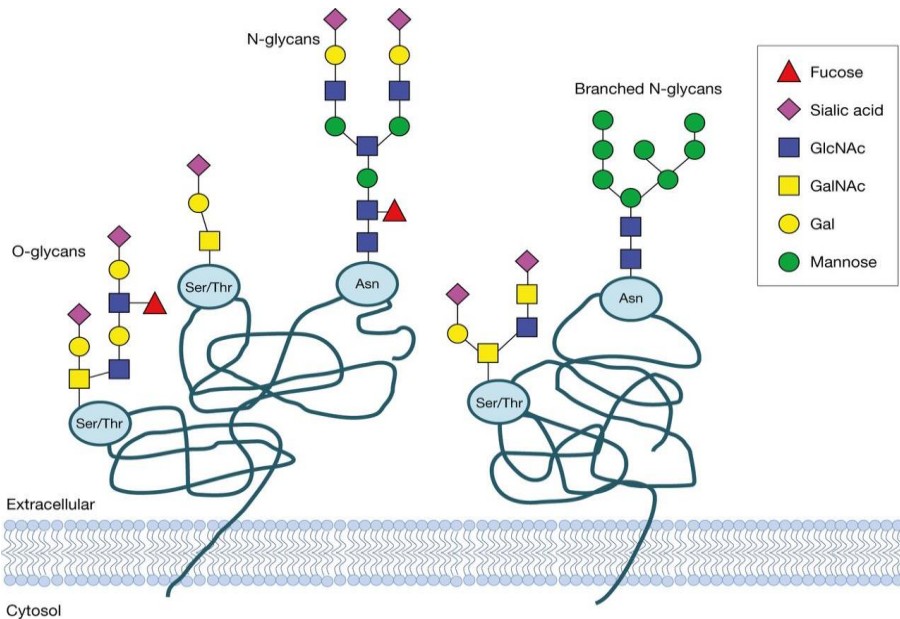

**Figure 1.** Visualization of the two major types of glycosylation: N-linked and O-linked glycosylation. Each symbol represents a specific monosaccharide. The most prominent sugar molecules comprise N-acetylglucosamine (GlcNAc), mannose (Man), N-acetylgalactosamine (GalNAc) and galactose (Gal) [12].

Various amounts of enzymes are required for each step during glycosylation, inducing diversity in the generated glycans. However, all proteins are N-glycosylated in the same way at first, and the diversity emerges only after trimming and glycan maturation in the

Golgi body. To diversify the glycans on individual glycoproteins, glycan processing in the Golgi apparatus comprises both cutting and adding sugars. However, the diversity of the N-glycans depends not only on the species and the type of cell but also on the physiological state of the cell, which can be changed, depending on the level of development and differentiation or disease [13,14]. The extent of glycosylation and the structures of the N-glycans have a decisive effect on the conformation, solubility and antigenicity of a protein, as well as its biological activity and half-life [15]. Since both the condition and the age of the cells and the cultivation conditions can cause unpredictable changes in the glycan spectrum, it is of great importance to understand the processes that lead to the heterogeneity to be able to influence the distribution of glycan in a targeted manner.

## 3. Procedure

The process of the N-glycosylation of proteins is a complicated form of biosynthesis that consists of a large network of reactions. The complexity of the N-glycan results from the variety and number of enzymes, their distribution on different levels of different cellular compartments, the concentration of the co-substrates and the many competitive reactions that are catalyzed by the enzymes. The biosynthesis of N-glycosylated proteins begins with the synthesis of a short oligosaccharide precursor in the membranous cisternae of the endoplasmic reticulum (ER). The oligosaccharides are covalently linked to the amino acid residue Asn (N) of the target proteins at the nitrogen atom on the side chain of the residue. The asparagine acceptor must normally be located in a consensus sequence of Asn-X-Ser/Thr, where X cannot be proline [16]. Between the two sequons, the glycosylation rate at the Asn-X-Thr sequon is much higher than that at the Asn-X-Ser sequon [5]. Recent studies have also revealed that the Asn-X-Cys sequon can be viewed as an infrequently used canonical glycosylation site as well [10,17,18]. The glycoprotein is then trafficked to the Golgi apparatus, where further addition and removal of sugar yields the highly variable complex oligosaccharide.

The Golgi apparatus is a dynamic organelle, consisting of a series of flattened membrane cisternae known as Golgi stacks. In addition to the flattened cisternae, each Golgi stack is surrounded by multiple transport vesicles, which are thought to carry the Golgi enzymes to the allotted compartments [19] (Figure 2, left). Various glycosyltransferases, glucosidases and nucleotide sugar transporters are located on the Golgi membranes, arrayed in a generally organized manner from the cis-Golgi to the trans-Golgi network, categorizing them into roughly five groups: cis, trans, medial, cis-Golgi network (cGN) and the trans-Golgi network (tGN). Each group is dependent on the specific substrate generated earlier in the pathway, allowing the glycans to be used as markers of passage through the Golgi apparatus [17]. The incoming cargo-loaded transport vesicles containing the immature proteins from the ER fuse with the Golgi apparatus at its cis-side [5,8]. N-glycans on the glycoproteins in the cis-Golgi are of the high mannose type, while hybrid and complex N-glycans are produced through the addition of GlcNAc, galactose, sialic acid and fucose sugars within the medial and trans-Golgi compartments [17,19]. Both hybrid and complex glycans can also carry a "bisecting" GlcNAc on the β-mannose of the core, which prohibits the additional addition of sugar on this particular GlcNAc. The subsequent processing steps can be divided into the addition of sugar on the core, the extension of the formed antennas and the termination of the extended antennas. This cycle of the addition and removal of sugar continues until the protein is correctly folded, when the vesicles carrying the matured proteins and lipids leave the Golgi apparatus from its trans faces [5].

Most cargo proteins move through the Golgi stacks in the anterograde direction, but a significant part of the transport also occurs in the retrograde direction inside the stack itself to ensure the retrieval of escaped proteins to their correct sites in the upstream cisternae or back to the ER [8]. However, how the transport generally takes place within the Golgi apparatus is a matter of dispute. There are two general models of transport: vesicular transport and cisternal maturation (Figure 2, right).

According to the cisternal maturation model, a cis cisterna is formed and progresses through the Golgi stack, gradually transforming as it accumulates medial and trans enzymes via vesicles that move from the later to the earlier cisternae. In the vesicular transport model, the enzymes in each cisterna remain unchanged, while the proteins move forward through the stack via vesicles that shift from earlier to later cisternae. Both models are considered plausible and could even potentially work together to fulfill the function of the Golgi apparatus.

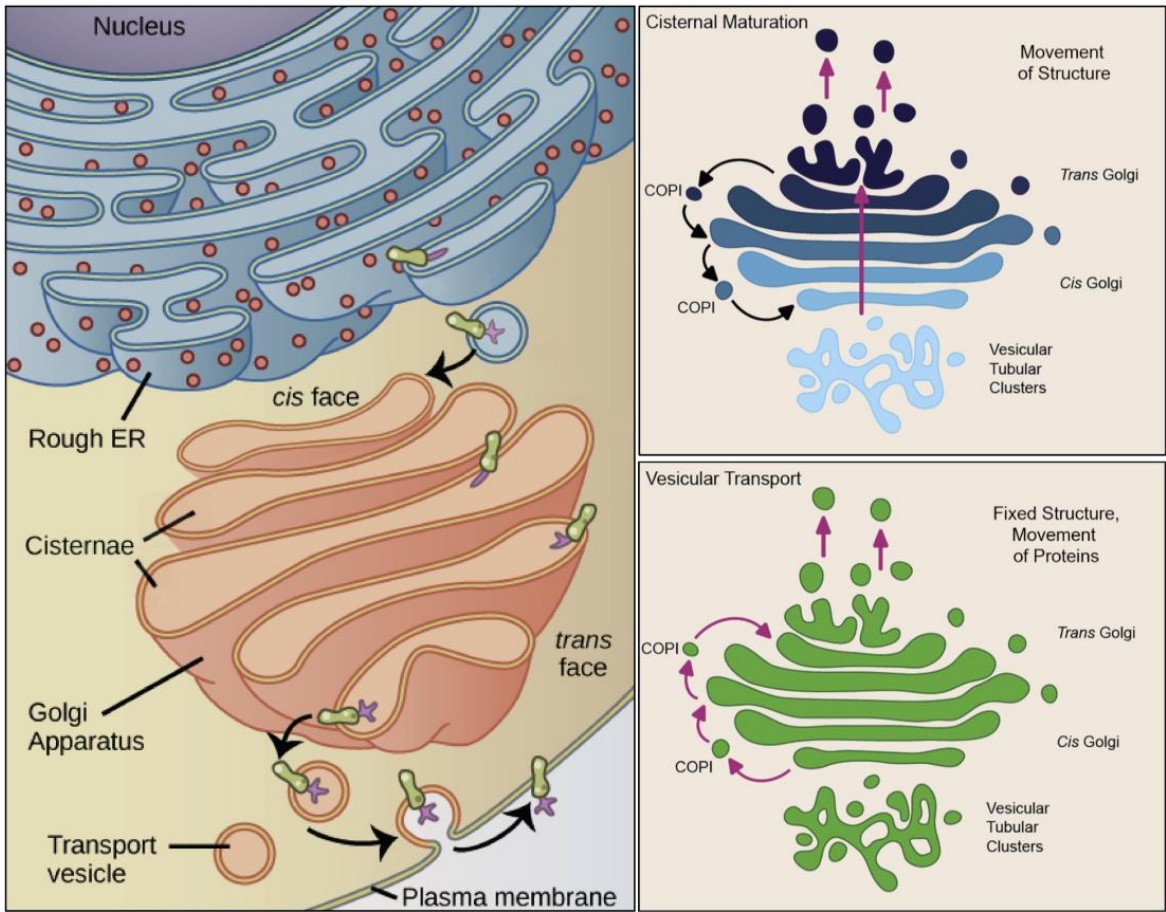

**Figure 2.** (**left**) N-glycosylation process of a protein, previously synthesized in the endoplasmic reticulum (ER), following its modification through the Golgi apparatus to its excretion from the cell [20]. (**right**) Two models of protein trafficking through the Golgi apparatus [21]. (**top right**) The cisternal maturation model states that the cisternae develop through a maturation process from the cis to the trans compartment and thus carry the cargo forward by their movement, while the Golgi enzymes are transported back to the earlier cisternae through COPI (coat protein Complex I) vesicles. (**bottom right**) In the vesicular transport model, it is assumed that the compartments are spatially fixed and that the cargo is transported from one cisterna to the next via COPI vesicles.

## 4. Agent-Based Modeling

Agent-based modeling (ABM) is an approach utilizing components of the system represented by autonomous, interacting entities called agents. At the simplest level, an agent-based model consists of a system of agents, the environment and the relationships between them. Agents are given attributes and initial rules of behavior that organize their actions and interactions. These rules may be either deterministic or stochastic in nature. The agents also can take on a typically finite collection of states. The agent's state depends on the agent's previous state, the state of the collection of other agents with which it interacts, and their interaction with the environment. The model is run multiple times and then

summarized to obtain a distribution of possible outcomes for the specified system [22]. The main outputs are the evolution of the system and its components, and a summary of the "final" state, which are useful for exploring and interpreting the system's behavior, structure and emergence [22]. ABM is effective for modeling systems containing spatial heterogeneity, wherein many agents follow the same set of rules [23]. A great advantage of agent-based modeling is the resulting spatial aspect, which can be traced back to local interactions and a heterogeneous environment. It is possible to program a large number of mobile agents that move over a grid of stationary agents ("patches"), potentially operating independently of one another and interacting with one another.

Computational advances have allowed the creation of a growing number of agent-based models across a variety of domains of application. The applications range from studying social interactions among individuals, discovering trends in the stock market, scheduling and improving the processes of a factory, exploring how cells react to drug treatments, predicting the spread of epidemics, and many other systems. At the molecular level, agent-based models are used for signaling pathways that are influenced by the spatial distribution and structural properties of the cell. They are also used to model metabolic reactions, but primarily at the multi-cell level [23,24].

### 4.1. Java for ABM

Describing agent-based simulation models in terms of objects is highly intuitive and leads to a natural implementation in OOP (object-oriented programming) languages, since they provide a natural way of expressing all of the different conceptual frameworks needed [25]. For this purpose, Java provides two indispensable features: classes and inheritance. A class is a user-defined blueprint from which objects are created. It represents the set of attributes or methods that are common to all objects of one particular type. Most object-oriented frameworks for agent-based modeling represent individual agents as instances of objects whose attributes correspond to the properties of the agents in the model [25]. In Java, it is also possible for one class to inherit the attributes and methods from another, allowing the storage of information in a hierarchical order. Inheritance is an important pillar of OOP and is an essential part of Java programming. The key to successfully creating a model for N-glycosylation was the development of a system for displaying and storing the glycan structure of each glycoprotein so that it could be accessed quickly and easily at any step, and it was possible to add or delete sugars in the structure dynamically.

### 4.2. Java Class Structure

In an agent-based model, various biological units are created that move within a defined space according to determined rules and can interact with one another. The overall system's dynamics are not defined in terms of a global function, but are rather the result of individuals' actions and interactions [26]. These interactions of the agents with the environment and with each other lead to a certain development of the model over time, which can be observed and evaluated. An agent is autonomous and self-directed, and can function independently of other agents and the environment. Each agent can be distinguished from every other agent by its attributes, typically a collection of variables. As an agent-based simulation progresses, the interactions of an agent with itself, other agents and the environment change the agent's state. In an agent-based simulation, agents can be defined at multiple levels, including individuals or groups of individuals, and there might be several different types or classes of agents. In this model, agents are differentiated between the protein class and the enzyme class, which both inherit mutual attributes and methods from the agent superclass.

The agent superclass (Figure 3) includes the properties needed by both the protein and enzyme subclasses, namely the name, their location in the environment and their partners, which are agents which share the same spatial proximity and are possible candidates to interact with. The location of the agents in the environment is saved in an integer array

consisting of their compartment (cis, medial, trans, tGN), the cisternae, lumen (lower membrane, center, upper membrane), and their respective x- and y-positions. Agents have rules for movement, for forming a complex with an enzyme, for reaction, for dissociation from the enzyme and for changing compartments.

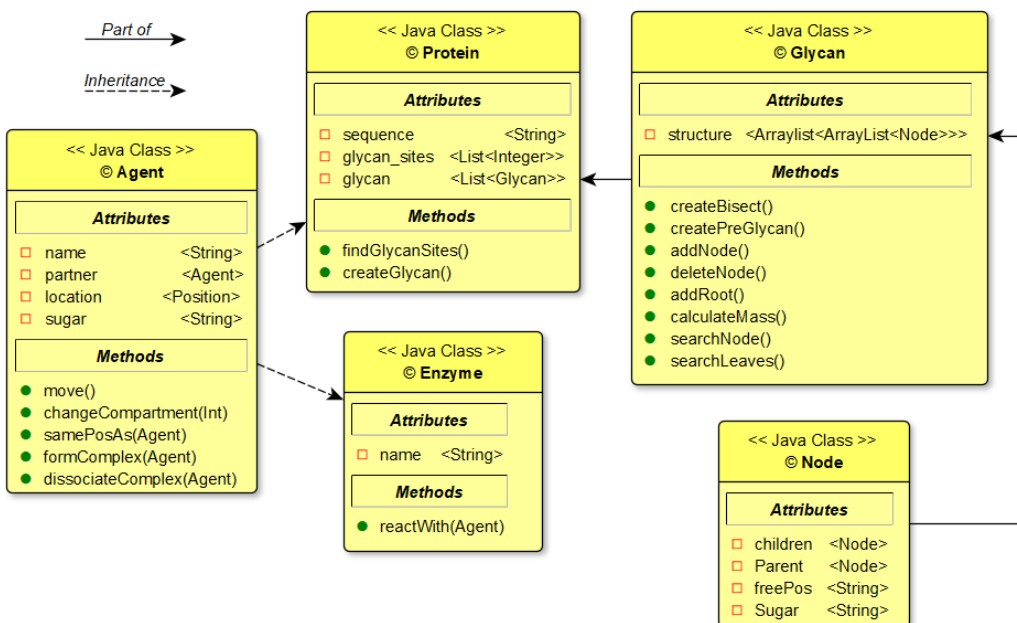

**Figure 3.** Class diagram of the agents for the simulation and their relationships.

The subclass representing the proteins inherits these attributes and also adds the amino acid sequence of the protein, the potential glycan binding sites and a list of glycans currently bonded to the protein. The protein sequence is searched for possible N-glycosidic bonds with an asparagine residue within the sequence of Asn-X-Ser/Thr/Cys, and their position in the sequence is saved as an integer. However, since only two-thirds of the consensus sequences are glycosylated, the Asn residues within the predicted consensus sequence only have a 66% chance of actually containing a glycan [11].

The enzyme class represents these glycosidases and glycosyltransferases that remove and add sugars from the proteins, respectively. It saves all possible reactions and the conditions of the enzymes with their specific substrate via the method *reactWith*.

### 4.3. Structure of the Glycans

The glycans themselves are saved as a class on their own containing their general structure, and several methods are used to analyze and search the glycans' structure. The composition of glycans is saved in the structure's attribute, which consists of the smallest structural unit in the model, the node class. The node class represents the monosaccharides, recording them in a nested array list to emulate their depth in the structure. The glycans' structure is constructed by adding the node class redundantly into a tree-like structure. The node objects are saved in a nested array list in the structure of the attributes of the glycans' class, indicating their depth in the glycans' composition (Figure 4). The different node objects are connected through their corresponding children and parent attributes in the node class, allowing a recursive approach to searching and analyzing the glycans' tree.

The node class includes the name of the sugar molecule, the available free positions for possible glycosidic bonds, and their overlying and underlying nodes (parents and children) in the glycan tree. The "children" attribute saves the underlying nodes as a hashmap with its glycosidic bond as the key, while also removing the bond from the freePos attribute. The freePos attribute shows the vacant sugar bonds available to other monosaccharides and is dependent on the nature of the node. A full table depicting all the available positions for

glycosidic bonds and the possible interactions between the monosaccharides is shown in Figure 5.

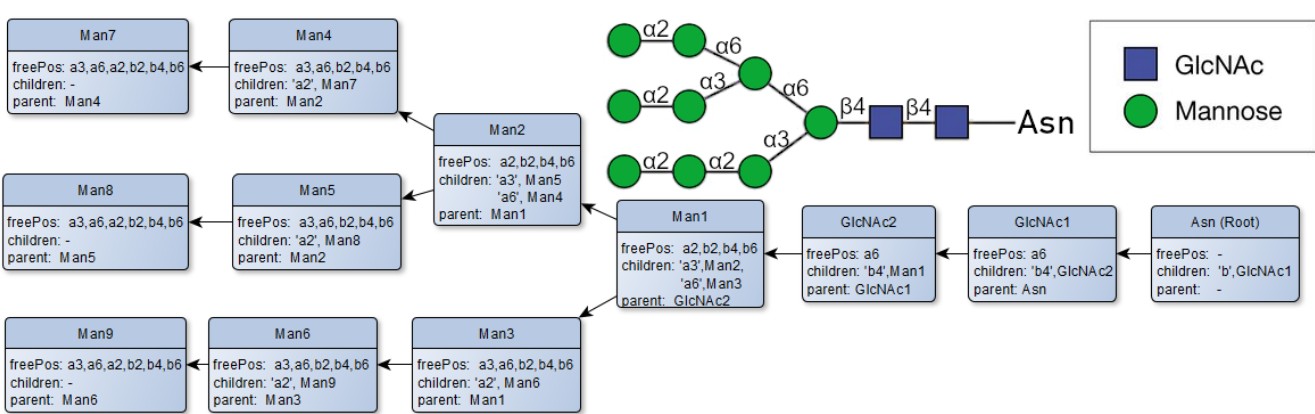

**Figure 4.** Example of a model representation of a glycan's structure saved in the glycan class.

Every glycan starts as a preglycan, as it has been synthesized at the ER and has been transported with the protein to the cis face of the Golgi body. In some cases, a mannose residue is removed from the glycans in the ER, resulting in a combination of Man8GlcNAc2 and Man9GlcNAc2 being transported to the Golgi. Therefore, if a protein with one or more glycans is created, each glycan has a 50% chance to either have a Man8GlcNAc2 or a Man9GlcNAc2 structure.

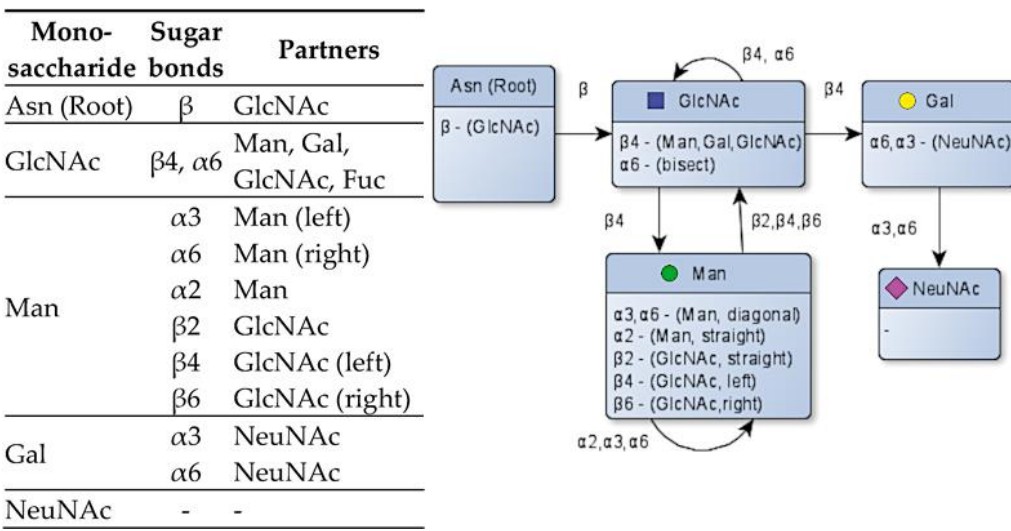

**Figure 5.** Species of monosaccharides and their potential bonding partners through the available sugar bonds [27] and a flow chart for the resulting process of glycosylation.

## 5. The Environment: The Golgi Apparatus

The environment is a global, passive element of the system where the agents interact with and is not considered an agent. The basis of the environment for our model is the conception of the Golgi apparatus. The cisternae of the Golgi apparatus vary in number, shape and organization in different cell types. The total number of cisternae is influenced by numerous factors such as growth, differential protein expression and cellular aging. Since the exact number of cisternae in the Golgi apparatus can vary, for the model, it is assumed that an average of eight cisternae are evenly distributed across the different compartments. To convert the Golgi apparatus into an environment for the agent-based simulation, its structure has to be converted into a three-dimensional model. For this purpose, the cisternae are considered individually, and their internal space is separated

into different levels within three dimensions: the lower membrane level, the lumen and the upper membrane level. For visualization, the three different levels are rotated by 90 degrees and are placed adjacent to each other to represent the cisternal structure as a two-dimensional grid (Figure 6). Since an enzyme has an average diameter of about 100 Å (10 nm), the size of a patch should correspond to 10 nm × 10 nm so that one enzyme fits into a patch.

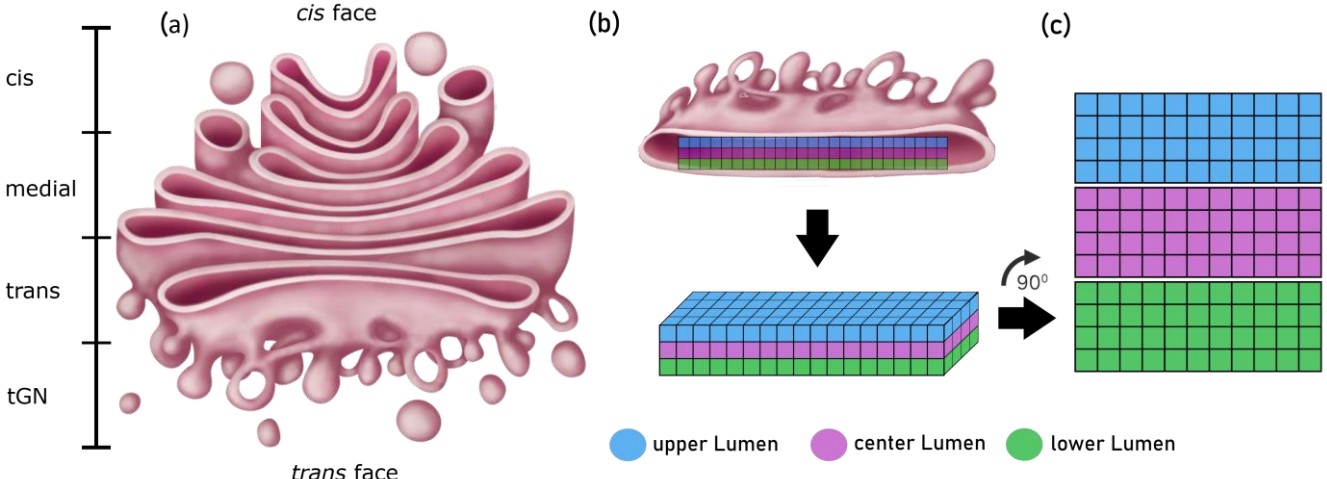

**Figure 6.** (**a**) Structure of the Golgi apparatus and its four distinct compartments. (**b**) Conversion of the cisternal structure of the Golgi apparatus into a 3-dimensional grid. (**c**) Two-dimensional presentation of the cisternal structure for visualization purposes.

After it has been determined how the Golgi should be structured in the model, its size has to be determined. Although many studies of the Golgi apparatus structure have been performed by light and electron microscopy, the full shape of the Golgi apparatus remains mostly unclear due to the technical limitations of the applied microscopy techniques. However, recent advances in microscopy technology, such as the use of high-power magnification, cryo-electron microscopy and 3D electron tomography, allow a general understanding of the dimensions and structures of the Golgi apparatus [27,28]. Microscopic images of the Golgi apparatus show an average diameter of 500 nm for a cisterna [29]. Assuming a discoid shape with a similar width would translate into a rectangular shape of 50 × 50 patches, for a total of 2500 patches for one level within a cisterna. The whole internal space of one cisterna would therefore consist of 7500 patches, while the entire Golgi structure would comprise 60,000 patches, resulting in a two-dimensional grid for the complete visualization of 60,000 = 150 × 400 patches (Figure 7).

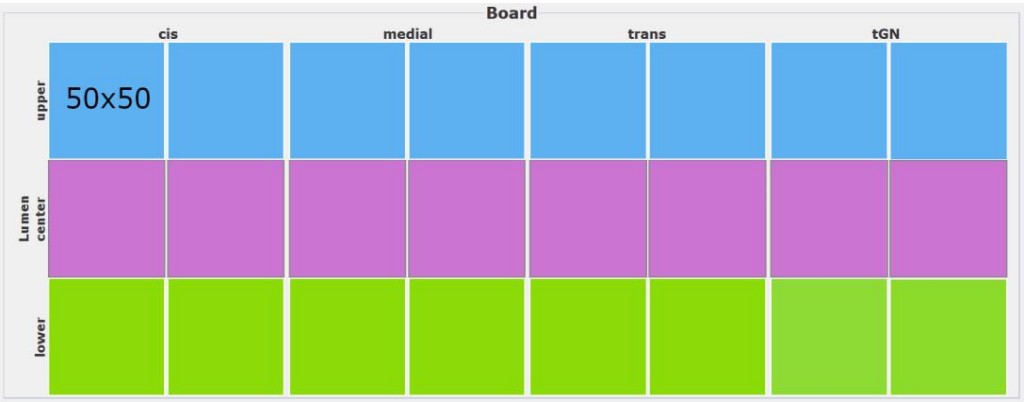

**Figure 7.** Two-dimensional visualization of the complete grid (150 × 400 patches in total) of the Golgi apparatus used in the model. The grid lines have been removed for better visibility.

*Enzymes*

The machinery of glycosylation includes numerous glycosyl transferases and glycosidases, as well as nucleotide sugars, which function as co-substrates. These enzymes are broad in scope, as glycosidic bonds have been detected on almost every functional group of proteins, and glycosylation has been shown to incorporate most of the commonly occurring monosaccharides to some extent. Glycosidases catalyze the hydrolysis of glycosidic bonds to remove sugars from proteins while enzymes that transfer mono- or oligosaccharides from donor molecules to growing oligosaccharide chains or proteins are called glycosyltransferases (Gtfs). For the simulation, the most fundamental enzymes for the glycosylation process in mammals [30] were selected and are listed in Table 1.

**Table 1.** List of all enzymes used in the simulation and their corresponding conditions for reacting [30].

| Enzyme | Condition | Reaction |
|---|---|---|
| **Man I** (α2-mannosidase I) mannosyl-oligosaccharid-1,2-α-mannosidase | Free α(1,2)-Man | Removes the α(1,2)-Man |
| **GnT I** (β2-GlcNAc-transferase I) α-1,3-mannosylglycoprotein-2-β-N-acetylglucosaminyltransferase | No α(1,2)-Man present | Adds a *β(1,2)*-GlcNAc to the α(1,3)-core-Man |
| **Man II** (α3/6-mannosidase II) mannosyl-oligosaccharid-1,3-1,6-α-mannosidase | Free α(1,3 and 6)-Man at α(1,6)-core-Man and GlcNAc at α(1,3)-core-Man; Inhibited by bisecting *β(1,4)-GlcNAc* | Removes α(1,3)-Man and α(1,6)-Man |
| **GnT II** (β2-GlcNAc-transferase II) α-1,6-mannosylglycoprotein 2-β-N-acetylglucosaminyltransferase | β(1,2) GlcNAc at α(1,3)-core-Man; Inhibited by bisecting β(1,4) GlcNAc; Inhibited, if Gal is present | Adds a *β(1,2)*-GlcNAc to the α(1,6)-core-Man |
| **GnT III** (β4-GlcNAc-transferase III) β-1,4-mannosylglycoprotein-4-β-N-acetylglucosaminyltransferase | β(1,2) GlcNAc at α(1,3)-core-Man; Inhibited, if Gal is present | Adds bisecting β(1,4) GlcNAc; |
| **GnT IV** (β4-GlcNAc-transferase IV) α-1,3-mannosylglycoprotein-4-β-N-acetylglucosaminyltransferase | β(1,2)-GlcNAc at α(1,3)-core-Man; Inhibited if β(1,4) Gal is at α(1,3)-Man-branch; Inhibited, if no Man or GlcNAc is at α(1,6)-core-Man; Inhibited by bisecting β(1,4)-GlcNAc | Adds a *β(1,4)*-GlcNAc to the α(1,3)-Man |
| **GnT V** (β6-GlcNAc-Transferase V) α-1,6-mannosylglycoprotein-6-β-N-acetylglucosaminyltransferase | β(1,2) GlcNAc at α(1,6)-core-Man Inhibited, if β(1,4) Gal is at α(1,6)-Man-branch; Inhibited by bisecting β(1,4)-GlcNAc | Adds a β(1,6)-GlcNAc to the α(1,6)-Man |
| **GalT** (β4-Gal-transferase) β-N-acetylglucosaminylglycopeptide-β-1,4-galactosyltransferase | Free GlcNAc | Adds β(1,4)-Gal to GlcNAc |
| **α6-FucT** (α6-Fuc-transferase) glycoprotein 6-α-L-fucosyltransferase | At least one GlcNAc added; Inhibited by bisecting β(1,4)-GlcNAc; Inhibited by a branch ending with Gal | Adds α(1,6)-core-Fuc |
| **SiaT** (α3/6-sialyl-transferase) β-galactosid-α-2,3/6-sialyltransferase | Free β(1,4)-Gal | Adds α(1,3)-NeuNAc to Gal |
| **α3-FucT** (α3-Fuc-transferase) glycoprotein 3-α-L-fucosyltransferase | β(1,4)-Gal at β(1,2)-GlcNAc | Adds antennary α(1,3)-Fuc |
| **GnT VII** (β3-GlcNAc-transferase VII) N-acetyl-lactosaminid-β-1,3-N-acetylglucosaminyltransferase | β(1,4)-Gal at β(1,2)-GlcNAc | Adds β(1,3)-GlcNAc to Gal |
| **GalNAcT II** (β4-GalNAc-transferase II) N-acetylneuraminylgalactosylglucosylceramide β-1,4-N-acetylgalactosaminyltransferase | α(1,3)-NeuNAc at β(1,4)-Gal | Adds β(1,4)-GalNAc to Gal |
| **GalNAcT III** (β4-GalNAc-transferase III) β-N-acetylglucosaminylglycopeptide β-1,4-N-acetylgalactosaminyltransferase | Free GlcNAc | Adds β(1,4)-GalNAc to GlcNAc |

The glycosylation enzymes are membrane-bound, which is why the glycans of the mobile glycoproteins can only be modified if they are in the vicinity of an enzyme. The precise spatial localization of the glycosyltransferases in the Golgi apparatus is of particular

importance for the formation of the different glycan structures. These enzymes are not segregated into clearly defined subcompartments but form overlapping concentration gradients across the stack [31]. The distribution of the enzymes over the compartments for the simulation is based on their intrinsic function and their spatial location in the glycosylation process. Mannosidase in the cis-Golgi regulates glycoprotein folding by removing the sugar residues in the dolichol-linked pre-glycan. In the medial portion of the Golgi apparatus, glycosyltransferases add sugar residues to the core glycan structure, creating the initial hybrid and complex N-glycans. Once this step has occurred, the majority of N-glycans are further trimmed by mannosidases, which are also present in the medial Golgi. The mature complex N-glycans are then created in the trans-Golgi by further sugar additions of galactosyltransferases and sialyltransferases. The distribution, as percentages, across the various compartments for the simulation is documented in Table 2.

**Table 2.** Distribution of enzymes in the Golgi compartments (compiled from [27]).

| Enzyme | Fraction in Compartment (%) | | | |
| --- | --- | --- | --- | --- |
| | Cis | Medial | Trans | tGN |
| Man I | 90 | 10 | 0 | 0 |
| GnT I | 20 | 50 | 30 | 10 |
| Man II | 10 | 50 | 30 | 10 |
| GnT II | 10 | 50 | 30 | 10 |
| GnT III | 10 | 50 | 30 | 10 |
| GnT IV | 10 | 50 | 30 | 10 |
| GnT V | 10 | 50 | 30 | 10 |
| GalT | 0 | 10 | 20 | 70 |
| SialT | 0 | 10 | 20 | 70 |
| α6-FucT | 10 | 50 | 30 | 10 |
| α3-FucT | 0 | 0 | 30 | 70 |
| GnTVII | 0 | 0 | 30 | 70 |
| GalNAcTII | 0 | 0 | 30 | 70 |
| GalNAcTIII | 0 | 0 | 30 | 70 |

## 6. Simulation Process

An agent-based simulation model depicts how the state of a system evolves over a duration of time. In any given discrete time period, the population of agents will follow a course of action given by the model, depending on its environment and other agents in its proximity. Time passes in discrete steps called "ticks". In the context of modeling, ticks are used to represent units of time. Any interactions that lead to changes in state within the same tick are considered to have occurred simultaneously [25]. The following procedures run sequentially after another within a tick:

1. Movement of the proteins;
2. Complex formation;
3. Reaction;
4. Dissociation.

### 6.1. Movement of the Proteins

The starting position of all proteins is a patch in the middle of the lumen plane of the cis compartment. A list of all the possible movements of a protein agent is seen in Figure 8.

There are seven possible movements for an agent in the simulation. It can either ascend or descend between planes, or move in a direction within a plane (forward, backward, left or right). There is also the possibility of an agent standing still. All seven movements have the same probability. If a movement results in crossing the defined border, it is treated as no movement and remains in place. It is also possible for more than one protein to be present in a patch. The enzymes are located at the membranes of the compartments and do not move. Movement is only possible if the protein agents have no bonded partner. In

order to analyze the movement of the proteins, the diffusion coefficient was calculated. A classic method for calculating the diffusion constant in Brownian motion is mean squared displacement (MSD) analysis. The coordinates of the proteins are measured at successive points in time, and the distance between these coordinates is calculated.

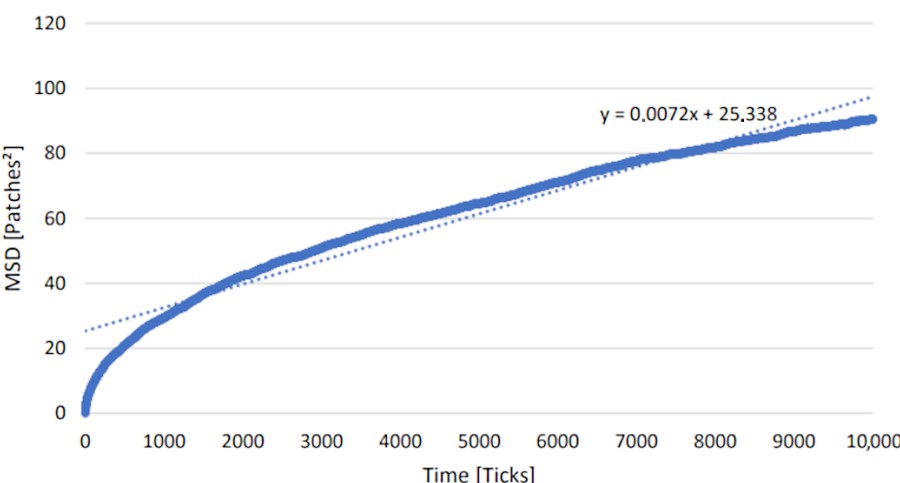

| Probability | Movement |
|---|---|
| 1:7 | No movement |
| 1:7 | Up |
| 1:7 | Down |
| 1:7 | Left |
| 1:7 | Right |
| 1:7 | Plane up |
| 1:7 | Plane down |

○ Protein – Agent  ● Possible Movement

**Figure 8.** Depiction of possible movements of a protein in a compartment. The present location is a white circle, while the possible positions after movement are grey circles. Movements leading outside the defined borders are treated as no movement.

A common method for determining the MSD is to use the Einstein relationship, which correlates the MSD with the diffusion coefficient and time. The MSD is plotted as a function of time, and a linear regression is carried out. The slope of the fit corresponds to the diffusion coefficient, characterizing the movement of the protein agents. For this purpose, a model has been established, in which the agents move on an infinite three-dimensional field with the same movement rules as in the glycosylation model. An average of 1000 proteins and 10,000 steps were chosen to calculate the distance traveled by the proteins. The resulting diffusion coefficient is 0.0072 patches$^2$/tick (Figure 9).

**Figure 9.** Mean squared displacement analysis for estimation of the diffusion coefficient. The movements of an agent are tracked over a certain period of time. The slope of the linear regression of the distance traveled relative to the starting point corresponds to the diffusion coefficient.

By modeling protein transport between the Golgi compartments, it was determined that the proteins would switch to the next compartment after a fixed number of ticks. The duration of the model was chosen so that the reactivity in all four compartments was as high as possible with the same residence time. Therefore, the overall reactions for 50 proteins in the first compartment were recorded over a period of 200,000 ticks (Figure 10). Of the maximum number of reactions, half of them occurred by the 20,000th tick. The residence time of the proteins per compartment was thus set to 20,000 ticks. Thereafter, a

protein could move to the next compartment if it was situated on the border to the next compartment and its movement was in direction to the next compartment. Since this can only happen if a protein is located on the upper membrane level of a compartment and the event of the compartment changing occurs with a probability of 5%, it can take a considerable number of steps until all glycoproteins have moved to the next compartment. The change into another compartment happens instantaneously in the period of one tick. By using this modeling concept, the simulation could depict the vesicular transport model, where each cisterna remains in one place with unchanging enzymes, and the proteins move forward through the stack via vesicles that move from earlier to later cisternae. By adjusting the modeling concept, it would also be possible to simulate the cisternal maturation model by blocking the movement of proteins between different cisternae and enabling the enzymes' movement.

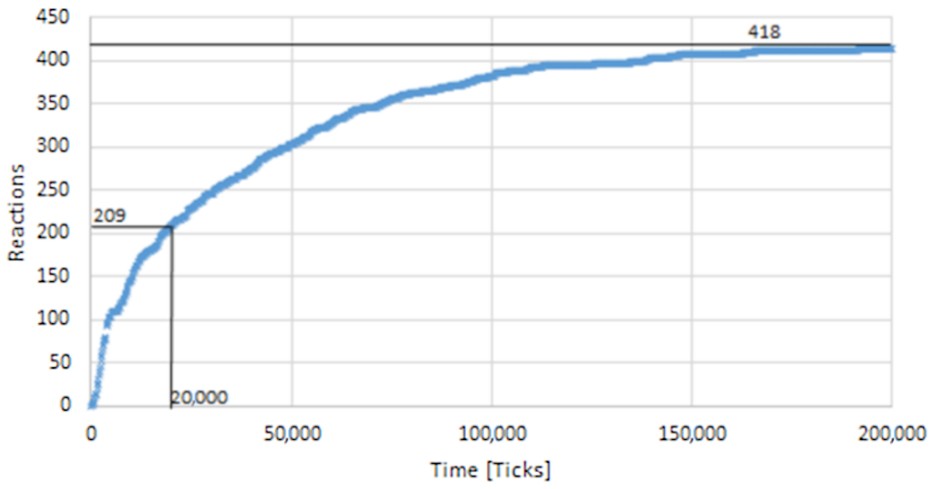

**Figure 10.** Sum of all reactions of 50 proteins over a time of 200,000 ticks in the cis compartment. With 50 proteins, a maximum of 418 reactions could be achieved. The number of ticks by which half of the maximum (i.e., 209 reactions) was reached is 20,000 ticks.

### 6.2. Complex Formation

The model integrates the probabilities of occurrence for the formation of a complex, and the reaction and dissociation of enzymes and proteins. These probabilities can be varied freely, resulting in different reaction rates. Thus, a possibility was created to transfer the differences between the reaction rates of different enzymes into the model. If a protein and an enzyme are in the same patch and have not yet formed a complex, they have a 50% probability of forming a complex. Every protein can bind to every enzyme, regardless of whether they are suitable reaction partners. If the following reaction procedure determines that they do not match, they dissociate again.

### 6.3. Reaction

In the reaction procedure, it is first determined which enzyme and protein have entered into a bond. The glycosylation enzymes are transmembrane proteins and therefore immobile elements. Their reactions can only take place when the glycoproteins are within the layer of the lumen adjacent to the membrane. Then the conditions for the enzyme are queried and checked for the protein. The reaction is performed if all the necessary conditions are met and a unique reaction probability for each enzyme is passed. Most reactions include either adding or removing a sugar moiety to an available location. A list of all the possible reactions can be seen in Table 1.

### 6.4. Dissociation

Three different dissociation procedures were defined: (1) if the protein is not a suitable substrate for the enzyme, both will move on without interaction; (2) the proteins dissociate after reacting with the enzyme, which happens in the same tick when the reaction takes place to avoid the protein reacting with the enzyme several times during a binding; (3) if the protein and enzyme are bound to one another but have not reacted, there is a 10% probability in every tick that they will nevertheless dissociate.

### 6.5. Simulation Output

At the end of the simulation, the structures of all the glycans and their corresponding proteins are saved. The resulting structures are saved into an XML format, a mass spectrometry file and a GlycoWorkbench2-compatible file for visualization, as seen in Figure 11. On average, the program was able to generate on average 28 unique structures after 50 repetitions for 50 proteins.

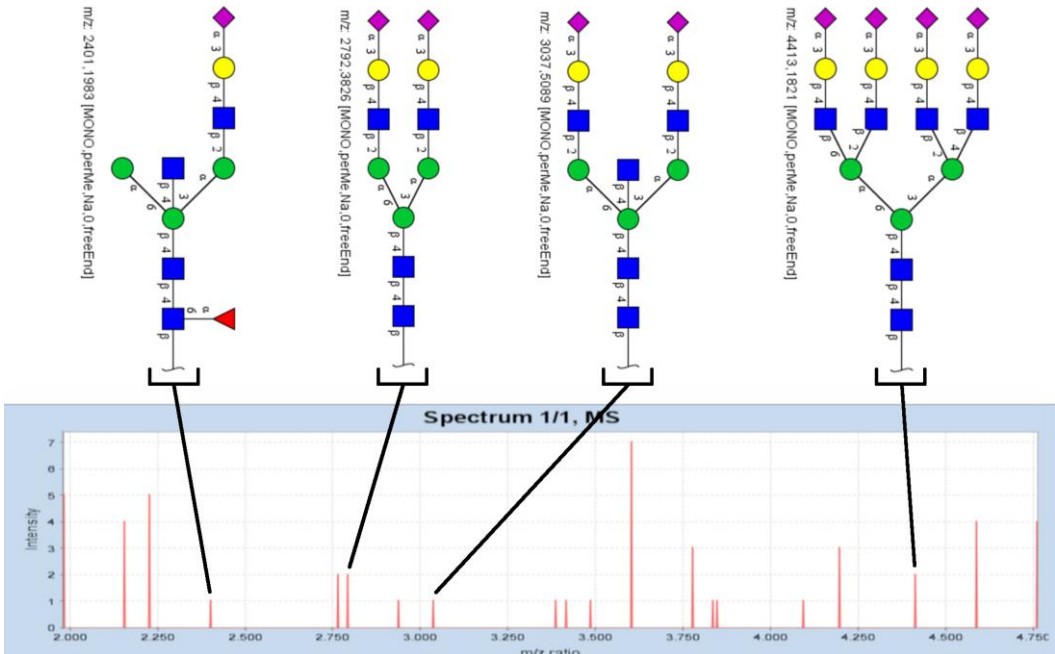

**Figure 11.** Glycan spectra generated by the simulation and visualized using GlycoWorkbench2. The monoisotopic mass ($m/z$ ratio) of the 20 resulting glycan structures and their relative abundance (Intensity) is shown as a peak in the mass spectrum plot. As examples, four of the 20 glycan structures are included in the figure.

### 7. Results

We developed a system for the representation of the glycans' structures (Figure 5), which resemble a tree-like structure, where every node represents a sugar moiety in the glycan tree. The coding of the glycans' structures was chosen so that it could be easily changed and expanded to enable the addition of further enzymes and sugars. In this way, the enzyme and sugar composition of the model could be adapted to a specific cell type that needs to be investigated. This is an important aspect for a possible later adaptation of the model for the production of a defined glycoprotein in a suitable expression system.

We successfully determined the potential value of the proof of concept for the simulation of N-glycosylation in the Golgi apparatus with an agent-based approach. The model shown can depict all steps of glycosylation in the Golgi apparatus. The protein agents carry N-glycans, move within the Golgi apparatus and encounter randomly distributed enzyme agents that are located on the Golgi membrane. After binding the protein, the enzymes can either transfer sugar to the glycan structure or split off a sugar, after which,

the complex dissociates again. A wide variety of reactions occur in the Golgi apparatus, and the glycan's structure changes continuously. After a certain period, the proteins can be transported to the next compartment. The source code and an executable file are available as Supplementary Materials.

## 8. Discussion

A model is an abstract and simplified representation of a given reality. However, if the model is too simplistic, the significance of the results might be distorted. For the sake of simplicity, assumptions have been introduced to and biological processes omitted from the model. For example, only the processing of the glycans in the Golgi apparatus has been considered. The transfer of the glycan precursor to the protein in the ER and the transport to the Golgi were not considered further. All glycoproteins are secretory proteins. It was assumed that the glycosyltransferases are saturated with nucleotide sugars and that only the glycan is necessary as a substrate for the reaction. It has also been observed that the Golgi apparatus is not a completely coherent organelle, but that the cisternae have compact and non-compact regions. The number of cisternae characteristic of the cell type could be of greater importance for modeling. The function of the various compartments has also not been fully clarified, e.g., it is assumed that the tGN is not an actual compartment but consists of several independent cisternae that are responsible for the further distribution of the proteins in the cell [32]. We still determined that the concept has value for application. To further increase the precision of the simulation, we identified several opportunities for improvement. The exact mechanism and time needed for the proteins to change cisternae or compartments need to be further analyzed to improve the movement between these components. Moreover, a limitation on the available agents on a single patch and the interactions between agent movements would make the conceptual approach more realistic. A better understanding of the correlation between time and ticks would lead to better synchronization of the simulation to the real-life occurrence of the processes. In subsequent research, we will integrate new concepts, demonstrating their feasibility and potential value.

One example for further work is that glycans on a mature glycoconjugate can have a versatile configuration, due to the dynamic and competitive character of glycan synthesis in the Golgi apparatus. A glycan may be a substrate for many different glycosyltransferases that compete, and a glycoconjugate may transit a Golgi compartment too quickly to react with all the enzymes capable of using it as a substrate. Additionally, glycan synthesis is also influenced by the Golgi apparatus' pH, the integrity of the peripheral Golgi membrane proteins, growth factor signaling, the Golgi membrane's dynamics and cellular stress, further increasing the complexity [17].

**Supplementary Materials:** The following supporting information can be downloaded at. https://www.mdpi.com/article/10.3390/fermentation9090849/s1. The source code of the software is available as Supplementary Materials. The executable can be run by using Java at the terminal "java -jar agent-based_glycosylation.jar".

**Author Contributions:** Conceptualization: P.G.; methodology: P.G. and C.J.; software: C.J.; validation: P.G. and C.J.; investigation: P.G. and C.J.; writing—original draft preparation: C.J.; additional writing, review and editing: P.G.; supervision, project administration and funding acquisition: P.G. All authors have read and agreed to the published version of the manuscript.

**Funding:** This research received no external funding.

**Institutional Review Board Statement:** Not applicable.

**Informed Consent Statement:** Informed consent was obtained from all subjects involved in the study.

**Data Availability Statement:** No new data were created or analyzed in this study. Data sharing is not applicable to this article.

**Acknowledgments:** We want to thank Stefan Hinderlich, Katja Karstens and Peter Neubauer for their support and advice. Special thanks to Susanne Fischer for the first proof of concept regarding the agent-based modeling of protein glycosylation. This study was financed by the Berlin University of Applied Sciences (BHT) from the doctoral scholarship program to promote cooperative doctorates.

**Conflicts of Interest:** The authors declare no conflict of interest.

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
