# Peer review of "Implementing an Agent-Based Modeling Approach for Protein Glycosylation in the Golgi Apparatus"

_fermentation, doi:10.3390/fermentation9090849_

Round 1

Reviewer 1 Report

The manuscript titled "An Integrated Model for Understanding Glycosylation Processes in the Golgi Apparatus" presents an agent-based modeling approach to simulate protein glycosylation in the Golgi apparatus. Modeling glycan biosynthesis is becoming increasingly important due to the far-reaching implications that glycosylation can exhibit, from pathologies to biopharmaceutical manufacturing. While the manuscript covers the most of the necessary aspects of the topic, some parts requires minor revisions:

1. While the manuscript provides an overview of the interaction between proteins and enzymes involved in glycosylation, it would be beneficial to further elaborate on the conditions required for complex formation, including the specific pH, temperature, and ionic strength. Discussing the factors that influence the dissociation of these complexes would enhance the understanding of the glycosylation process.

2. Standard formation of simulation model should go through essential steps to ensure its accuracy and reliability - model development, parameterization, validation, and verification. While first two steps: model development (defining the agents, their behaviors, and the rules of interactions) and parameterization (assigning values to the parameters of the model based on experimental data or literature) are satisfied, validation (comparing the model's output to experimental data to confirm that it correctly represents the modeled system) and verification (checking the model's consistency and accuracy) parts should be enhanced to make the model more successful.

3. Numerous technical mistakes should be corrected after a thorough critical reading. Examples are:

a) repeated textual segments (for instance, within the text, it is repeated thrice that Golgi apparatus consist of four distinct compartments  four distinct compartments namely cis, medial, trans, and tGN; and twice that Golgi apparatus is a dynamic organelle, consisting of series of flattened membrane cisternae, known as the Golgi stacks)

b) not all components of the grahics are specifically indicated in graphic legends

c) diverse abbreviation modalities were used for same context ( N-X-S/T, N-X-S/T/C, Asn-X-Ser/Thr/C, Asn-X-100 Ser/Thr)

d) referencing in the abstract

4. Do you possess the permission for utilization of figures from the manuscripts you've referenced in the figure caption? Using others' figures without permission in this manner is not allowed.

Based on all that has been mentioned, I suggest accepting the paper for publication with minor revisions.

I find the English language satisfactory.

Reviewer 2 Report

- Figure 4: Authors need to add a legend to the scheme.

- Line 255: Is it possible using this model to predict and confirm which structure, Man8GlcNAc2 or a Man9GlcNAc2, is formed?

- Lines 286-289: What about the high magnification, cryo-EM, 3D Electron Tomography, ........ as techniques to visualize the Golgi apparatus structure?

- Lines 356 and 358; The authors need to detail more how the probabilities of 4:7 and 2:4 happened for the possible movements of  the proteins in a compartment.

- Move Figure 8 under the line 360.

- Line 390: lowercase "P" of "Protein".

- Does this model can be applied too for prokaryotic organisms for the simulation of the O-glycosylation ?

- References to be quoted:

    - DOI: 10.1111/j.1600-0854.2004.00186.x

   - DOI: 10.1007/978-1-0716-2639-9_12

Reviewer 3 Report

Jetschni, C, Gotz, P., et al., et al have discussed emphasizing the significance of glycoproteins in vital biological processes and their roles in regulating activities and signaling pathways. The complexity of enzymes and resulting glycoproteins makes simulating these processes challenging.

The author focused on the issue of creating a simulation of N-glycosylation, particularly within the Golgi structure, using an agent-based model, defining interactions and movement patterns among proteins and enzymes, treating them as agents, whereas the Golgi structure is transformed into a grid with specific enzyme distributions and the resulting structures are saved in various file formats like mass spectrometry file, and a GlycoWorkbench2 compatible file for visualization.

In this study, the authors demonstrated and developed a system to represent glycan structures resembling tree-like arrangements, with each node representing a sugar moiety in the structure. The coding of these structures is designed for flexibility, allowing easy modifications and expansions to include additional enzymes and sugars. This adaptability is crucial for tailoring the model to specific cell types and investigating their enzyme and sugar compositions. This adaptability is especially useful for potentially producing specific glycoproteins inappropriate expression systems.

The authors have also shown that their proof-of-concept simulation for N-glycosylation in the Golgi apparatus using an agent-based approach yielded promising results. The model effectively captures all stages of glycosylation within the Golgi apparatus. Proteins, acting as agents, carry N-glycans and move within the Golgi, encountering enzyme agents distributed on the Golgi membrane. Upon binding, these enzymes can either add or remove sugars from the glycan structure before dissociating from the complex. The Golgi undergoes various reactions, leading to continuous changes in the glycan structure. Eventually, the proteins can be transported to the next compartment after a certain duration. This model showcases the potential value of the approach in simulating N-glycosylation in the Golgi apparatus.

However, the concerning part of this study is that the predicted model serves as a simplified and abstract depiction of a particular reality, making it potentially distorting the outcomes, as well as certain assumptions have been incorporated while omitting certain biological processes in the model, which was also mentioned by the authors.

The novelty of this study lies in its approach to simulating the N-glycosylation process within the Golgi structure using an agent-based model using simulating N-glycosylation in silico. And transforming the Golgi structure into a grid with compartmentalized enzyme distributions.

Nonetheless, the article seemed to possess a few major concerns, especially regarding figure description and modeling. Overall, the clarity of the text is good and easily understandable but a few of the points need to be discussed properly with references and schematic diagrams.

The manuscript has very few typographical and grammatical errors which need to be corrected. In general, the manuscript can accomplish the caliber of quality for consideration for publication in the Journal “Fermentation”. The authors are advised to consider the comments below:

Comments

1.      Page 2 / Line 49 / “process” - The singular noun process follows a number other than one. Consider changing the noun to the plural form.

2.      Page 6 / line 213 / “position” - It seems that position may not agree in number with other words in this phrase.

3.      Page 8 / line 262/  “apparatus” - The word apparatus should be capitalized in this context.

4.      Page 10 / line 319 / “are” - The plural verb are does not appear to agree with the singular subject Mannosidase. Consider changing the verb form for subject-verb agreement.

5.      Page 12 / line 378 / Change “The average” to “An average” - It seems that there is an article usage problem here.

6.      Abstract / Please avoid using references in the abstract.

7.      Introduction / first paragraph / Please provide appropriate references to support the statement – Engineered N-linked glycosylation into genes of interest, leading to glycosylated recombinant proteins, antibodies, peptides, and fusion proteins, which improves the stability of these molecules and expands their potential clinical applications.

8.      Figure 1 / Please include a few more details for better clarification of the image – Subdivision of N-linked glycans (oligomannose, hybrid, complex)

9.      Figure 2 / The schematic diagram is incomplete on various levels. Some structures are not marked or described in the figure legends. Especially the “vesicular transport “ image needs to be redone in a proper manner.

10.   N-glycosylation of proteins consists of a large network distributed in different cellular compartments (mentioned in Part -3) – Please provide details of the involvement of the endoplasmic reticulum with proper references and schematics.

11.   Figure 5 – the description of the implemented monosaccharides and their relationship through their potential sugar bonds is not properly done. Please provide appropriate references and text supporting this image.

12.   The table describing the list of all Enzymes used in the simulation and their corresponding conditions for reacting – have no reference panel. Please add appropriate references.

13.   Check the lettering of the “Table:2”

14.   Write a short explanation of Figure 9 in its figure legend.

Overall, the clarity of the text is good and easily understandable. The manuscript has very few typographical and grammatical errors which need to be corrected.
